# Training Language Model Agents to Find Vulnerabilities with CTF-Dojo

Terry Yue Zhuo [* 1 2]   Dingmin Wang [2]   Hantian Ding [2]   Varun Kumar [2]   Zijian Wang [* 3]

## Abstract

Large language models (LLMs) excel when trained in executable runtime environments that provide verifiable feedback, especially for software engineering tasks. However, scalable and general-purpose execution-grounded environments remain limited. We introduce CTF-DOJO, the first large-scale executable runtime designed for cybersecurity agent training with verifiable feedback, featuring 658 reproducible, Dockerized Capture-The-Flag (CTF) challenges. To support rapid scaling, we present CTF-FORGE, an automated pipeline that transforms public artifacts into executable environments in minutes, eliminating extensive manual setup. Training on just 486 execution-verified trajectories from CTF-DOJO yields up to 11.6% absolute gains over strong baselines on *InterCode-CTF*, *NYU CTF Bench*, and *Cybench*. Our best 32B model achieves 31.9% Pass@1, comparable to DeepSeek-V3-0324 and Gemini-2.5-Flash. By framing CTF-style tasks as resources for executable-agent learning, CTF-DOJO demonstrates that execution-grounded training signals are not only effective but effective for improving high-performance cybersecurity agents without dependence on costly proprietary systems. We have release the codebase at https://github.com/amazon-science/CTF-Dojo.

## 1. Introduction

Advanced cybersecurity necessitates the ongoing analysis of increasingly complex software systems. As globally connected infrastructures expand, their attack surfaces expand as well, making traditional manual security analysis insufficient for timely vulnerability identification and remediation.

This urgency has spurred major research efforts, such as the DARPA Cyber Grand Challenge (Song & Alves-Foss, 2015) and DARPA AIxCC (DARPA, 2024), which focus on building autonomous systems capable of discovering and validating software flaws. In this context, Capture The Flag (CTF) competitions have emerged as the de facto benchmark for evaluating the cybersecurity reasoning abilities of machine learning models, demanding advanced, multi-step adversarial strategies to uncover system vulnerabilities and retrieve hidden flags (Anthropic, 2025a; xAI, 2025; OWASP GenAI Project (CTI Layer Team), 2025).

Previous works have demonstrated promising results in applying large language model (LLM) agents to CTF challenges (Hurst et al., 2024; Jaech et al., 2024; Anthropic, 2025b; Abramovich et al., 2025), with systems like ENIGMA (Abramovich et al., 2025) achieving substantial progress on complex security tasks. While these approaches enable frontier proprietary models to achieve strong performance, they fail short when applied to open-source LLMs due to the lack of agentic training data. Recently, Zhuo et al. (2025) shows that training on thousands of synthetic agent trajectories can close the gap between proprietary and open-source LLMs. However, synthesizing a large number of long-horizon trajectories from teacher models requires substantial computational resources, limiting generalization under budget constraints. Moreover, the validity of synthetic trajectories is hard to verify without runtime environments, limiting their reliability for training in high-stakes, safety-critical domains.

To address these limitations, we present CTF-DOJO, a large-scale execution environment that contains hundreds of fully functional CTF challenges in secure Docker containers. CTF-DOJO leverages CTF artifacts (e.g., challenge descriptions and files to reproduce each challenge) from pwn.college, a public archive developed by Arizona State University for hands-on cybersecurity education, now used in 145 countries and actively maintained by a team of professors and students. However, setting up the runtime environment for CTF challenges is extremely difficult for non-professionals and can take up to an hour per task even for experienced practitioners (documented Section 2). To eliminate this bottleneck, we propose CTF-FORGE (Figure 1), an automated pipeline that leverages LLMs to create hundreds of Docker images for CTF-DOJO within minutes,

---

[*]: Work done during an internship at Amazon. [1]Monash University [2]AWS AI Labs [3]Meta Superintelligence Labs. Correspondence to: Terry Yue Zhuo <terry.zhuo@monash.edu>.

*Proceedings of the $43^{rd}$ International Conference on Machine Learning*, Seoul, South Korea. PMLR 306, 2026. Copyright 2026 by the author(s).

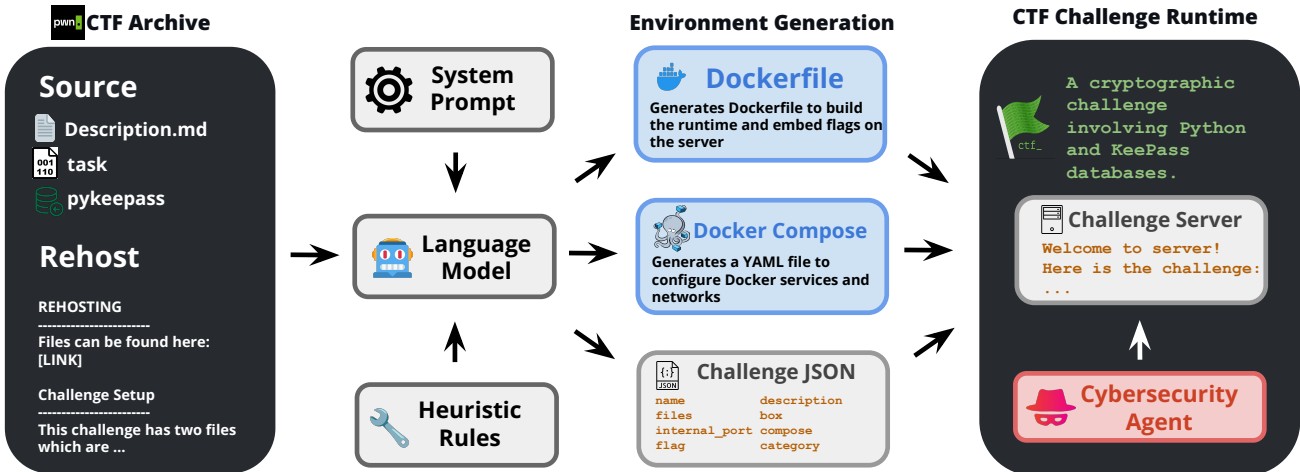

*Figure 1.* CTF-FORGE powers automated creation of configuration files from publicly sourced CTF artifacts for containerizing CTF challenges.

achieving over 98% success rate through manual validation.

During trajectory collection from multiple LLMs within CTF-DOJO, we found that weaker models struggle to solve CTF challenges independently (detailed in Section 4.1). To improve yield rates, we collect diverse CTF writeups from CTFtime[1] and incorporated them as inference-time hints. Although we notice that only 23% of the CTF-DOJO challenges matches at least one writeup, we empirically find that such writeup content, when available, can significantly boost the success rate of LLMs up to 64% relatively gains. Notably, while building these environments, CTF-DOJO uncovered four bugs from the existing `pwn.college` collection[2].

Models trained on CTF-DOJO trajectories achieve superior performance on over 300 tasks across three established CTF benchmarks. Through the extensive analysis, we identify three key findings for building effective cybersecurity agents: (1) writeups are crucial for training, particularly when working with data generated by weak models, (2) augmenting the runtime environment (e.g., server domains and flags) helps models yield more solved more CTF challenges, and (3) employing diverse teacher LLMs in CTF-DOJO leads to better task diversity and stronger performance. We hope our insights from the proposed CTF-DOJO can shed light on the future development of cybersecurity agents. Our work provides following contributions:

- We introduce CTF-DOJO, the first large-scale, execution-ready environment for cybersecurity agent training, offering hundreds of verified CTF challenges in isolated Docker containers.

- We propose CTF-FORGE, a scalable pipeline that leverages LLMs to automate the generation of Docker-based runtime environments, achieving over 98% success rate through manual validation.

- We conduct thorough analysis through extensive ablation studies, identifying key factors that influence agent performance, including the presence of hint-guided trajectory collection, runtime environment augmentation, and teacher model diversity.

## 2. CTF-DOJO: Environment for Building Powerful Cybersecurity Agents

CTF-DOJO is the first environment designed to synthesize verified agent trajectories for training LLMs on offensive cybersecurity tasks involving vulnerability detection and exploitation. As shown in Table 1, existing cybersecurity execution environments either lack agentic task instance or are not designed for training purposes, creating a critical gap in the development of capable security agents. Inspired by the success of trajectory-based learning in software engineering agents (Jimenez et al., 2024; Yang et al., 2024a), CTF-DOJO adapts this paradigm to cybersecurity by sourcing publicly available CTF artifacts and transforming them into executable and interactive environments.

Different from prior pipelines for software engineering tasks (Pan et al., 2024; Xie et al., 2025; Yang et al., 2025b), which often require human effort or complex multi-agent systems to construct Docker environments, our approach is lightweight and fully automated. Towards that end, we introduce CTF-FORGE, a pipeline that automatically builds Docker containers for CTF-DOJO. While manual setup can

---

[1] https://ctftime.org/
[2] We have filed issues in their official repository.

take up to an hour per challenge even for experts[3], CTF-FORGE completes each container in 0.5 seconds on average, reducing weeks of total setup time to just minutes.

## 2.1. Source Data Collection

We begin by surveying CTF collections that offer diverse challenges from CTF competitions. During our initial exploration, we determine a few candidates: (1) Sajjadium's CTF Archives[4], (2) r3kapig's Notion[5], (3) CryptoHack CTF Archive[6], (4) archive.ooo[7], and (5) `pwn.college`'s CTF Archive[8]. However, most of these collections suffer from inconsistent maintenance, lack standardization across challenge formats, or are limited to specific categories (e.g., CryptoHack focuses solely on cryptography). We determine that `pwn.college`'s CTF Archive is not only free of these issues but additionally provides brief information about the steps to reproduce each CTF challenge. Table 2 shows the distribution of 658 CTF challenges (as of 2025/07) after decontaminating any tasks from evaluation benchmarks, demonstrating the diversity of CTF instances across different categories and competition events hosted between 2011 and 2025. We perform decontamination by matching competition names, competition years, and individual challenge names between the pwn.college CTF Archive and all evaluation benchmarks, followed by manual verification of matched candidates. Using this process, we identify and remove 3 overlapping challenges from the training data. While this procedure cannot fully eliminate broader semantic similarities between CTF tasks, it substantially reduces the risk of direct benchmark contamination.

CTF challenges employ two primary flag-handling mechanisms. The first type uses predefined flags, hashed with SHA-256 and verified through a provided binary executable (e.g., `flagCheck`) that confirms submission correctness. Since these flags were manually captured and encoded, they are subject to occasional errors (see 4 identified bugs in Appendix G). The second type relies on dynamic flag generation, where the correct flag is generated at runtime and stored in a system path such as `/flag`. In those challenges, participants must verify the system during execution to retrieve or compute the correct flag, rather than match against a static value.

---

[3]This has been attempted by one of the authors.
[4]https://github.com/sajjadium/ctf-archives
[5]https://r3kapig-not1on.notion.site
[6]https://cryptohack.org/challenges/ctf-archive/
[7]https://archive.ooo/
[8]https://github.com/pwncollege/ctf-archive

## 2.2. CTF-FORGE: Automatic Environment Creation for CTF Challenges

Figure 1 illustrates CTF-FORGE, a pipeline employing DeepSeek-V3-0324 (Liu et al., 2024) to generate environments and metadata for CTF runtime. After we source the CTF artifacts from `pwn.college`'s CTF Archive, we design a set of prompts to instruct LLMs to generate the compulsory files for Docker images in multiple stages. First, we determine whether the CTF challenge requires a containerized server to interact with. Such servers are typically needed for web challenges, binary exploitation challenges, and cryptography challenges that provide interactive services. The pipeline automatically detects server requirements by analyzing the presence of flag verification files (SHA256 checksums or check scripts) and challenge descriptions. For existing CTF runtime, we can categorize them into several challenge types: 1) Web challenges that require web servers (Apache/Nginx) to serve PHP, Python, or Node.js applications; 2) Binary exploitation challenges that need socat to host binary services on port 1337 with appropriate library dependencies; 3) Cryptography challenges that may require Python runtime environments for cryptographic services; 4) Reverse engineering challenges providing downloadable binaries and potentially analysis services; and 5) Forensics challenges offering evidence files for offline analysis. The pipeline employs category-specific guidelines and adaptive Docker setup strategies to handle different architectures (32-bit vs 64-bit), library dependencies, and runtime environments. For each challenge type, CTF-FORGE generates appropriate Dockerfiles with proper base images, package installations, file copying, and service configurations, then produces `docker-compose.yml` files for orchestration and `challenge.json` metadata files that describe the challenge structure and provide flag verification mechanisms.

## 2.3. Building Sustainable Environment for Cybersecurity Agents

To ensure CTF-DOJO serves as a robust foundation for long-term research on autonomous cybersecurity agents, we emphasize sustainability across two dimensions: reliability and scalability.

**Reliability** To ensure the reliability of the CTF environments created via CTF-FORGE, we implement an automated validation script that performs two critical checks: (1) whether the Docker containers can be successfully built and executed without errors, and (2) whether the CTF services inside the containers respond correctly to network communication on the expected ports. We run CTF-FORGE three times independently on all 658 CTF challenges to evaluate consistency and determinism. Across these runs, 98% (650) of the challenges consistently pass all checks,

*Table 1.* CTF-DOJO is the first cybersecurity executable environment deriving agent trajectories for training. *Detection:* whether the task requires vulnerability detection; *exploitation:* whether the task needs LLMs to verify the detected vulnerabilities; *Agentic:* whether each instance is equipped with an interactive environment for exploitation; *Real Task:* whether each instance reflect real-world development.

| Executable Environment | Detection | Exploitation | Agentic | Real Task | # Total | # Train |
|---|---|---|---|---|---|---|
| SecRepoBench (Dilgren et al., 2025) | ✗ | ✗ | ✓ | ✓ | 318 | 0 |
| CVE-Bench (Wang et al., 2025a) | ✗ | ✗ | ✓ | ✓ | 509 | 0 |
| CVE-Bench (Zhu et al., 2025) | ✗ | ✓ | ✓ | ✓ | 509 | 0 |
| SEC-bench (Lee et al., 2025) | ✗ | ✓ | ✓ | ✓ | 200 | 0 |
| CyberGym (Wang et al., 2025b) | ✗ | ✓ | ✓ | ✓ | 1,507 | 0 |
| CyberSecEval 3 (Wan et al., 2024) | ✓ | ✓ | ✓ | ✗ | 6 | 0 |
| SecCodePLT (Yang et al., 2024b) | ✓ | ✓ | ✓ | ✗ | 1,345 | 0 |
| InterCode-CTF (Yang et al., 2023) | ✓ | ✓ | ✓ | ✓ | 100 | 0 |
| NYU CTF Bench (Shao et al., 2024) | ✓ | ✓ | ✓ | ✓ | 200 | 0 |
| Cybench (Zhang et al., 2025b) | ✓ | ✓ | ✓ | ✓ | 40 | 0 |
| BountyBench (Zhang et al., 2025a) | ✓ | ✓ | ✓ | ✓ | 40 | 0 |
| CTF-DOJO (Ours) | ✓ | ✓ | ✓ | ✓ | 658 | 658 |

*Table 2.* Challenge distribution across CTF datasets.

| Benchmark | Level | # Competition | # Crypto | # Forensics | # Pwn | # Rev | # Web | # Misc | # Total |
|---|---|---|---|---|---|---|---|---|---|
| *Training* | | | | | | | | | |
| CTF-DOJO | Multi-Level | 50 | 228 | 38 | 163 | 123 | 21 | 85 | 658 |
| *Evaluation* | | | | | | | | | |
| InterCode-CTF | High School | 1 | 16 | 13 | 2 | 27 | 2 | 31 | 91 |
| NYU CTF Bench | University | 1 | 53 | 15 | 38 | 51 | 19 | 24 | 192 |
| Cybench | Professional | 4 | 16 | 4 | 2 | 6 | 8 | 4 | 40 |

demonstrating high reliability of the pipeline in producing stable, executable environments for cybersecurity agents. Additionally, we sample 10% of the built CTF tasks and manually run the executables within each runtime to verify expected behavior including flag submissions.

**Scalability** While CTF-DOJO currently contains fewer instances than existing software engineering environments that covers thousands of instances (Pan et al., 2024; Xie et al., 2025; Yang et al., 2025b), each CTF challenge environment is uniquely designed, mimicking diverse real-world software systems rather than variations of a single codebase that is common in SWE tasks. To enhance scalability over time, CTF-DOJO builds on the actively growing CTF collections from the `pwn.college` community. As new challenges are added, CTF-FORGE can continuously and automatically convert them into interactive environments with minimal manual effort, enabling CTF-DOJO to scale organically alongside community-driven CTF development.

### 2.4. Training Data Construction

We introduce a data pipeline to produce a large corpus of high-quality, multi-turn interaction traces from CTF-DOJO. This process supports the development of CTF-solving agents that require diverse, realistic demonstrations of iterative security problem-solving behavior.

**Agent Scaffold** We build on ENIGMA+ (Zhuo et al., 2025), a recently introduced agent scaffold designed for scalable and consistent evaluation of agents on cybersecurity tasks. ENIGMA+ extends the original ENIGMA framework to better support cybersecurity environments by incorporating interactive tools for debugging and remote server interaction. Notably, ENIGMA+ improves evaluation efficiency by executing tasks in parallel using isolated Docker containers, reducing runtime from days to hours for large-scale experiments. It also enables the control of agent interactions based on the number of interaction steps (e.g., 40 turns) rather than monetary cost, which aligns with best practices in agent evaluation. Additionally, it replaces ENIGMA's context-heavy summarization module with a lightweight alternative better suited for binary analysis outputs. Within this scaffold, we integrate the CTF-DOJO environment and collect agent trajectories through structured interactions.

**Trajectory Collection** Within the ENIGMA+ scaffold, we deploy Qwen3-Coder and DeepSeek-V3-0324 to attempt solving CTF challenges in CTF-DOJO with a temperature of 0.6, top-p of 0.95, and rollout count of 6. For each challenge instance, the agent is given the original task description and interactive access to the containerized environment, capped at 40 turns. We log every system command, intermediate output, and reasoning step until either the flag is captured

or the turn budget is exhausted. Successful trajectories are stored in structured JSON format for downstream filtering and training. Our initial large-scale runs reveal that many trajectories stall due to brittle exploitation strategies or failure to discover the correct toolchain. While some challenges yield multiple successful runs, a large fraction remain unsolved or are solved only rarely, leading to a skewed dataset concentrated on limited tasks.

**Inference-Time Bag of Tricks** To increase the yield rate of successful trajectories on CTF challenges, we introduce two inference-time techniques (analyzed in Section 4). *First, we leverage publicly available CTF writeups to provide task-specific hints to LLMs*. Specifically, we collect 8,361 writeups and apply fuzzy matching to align them with challenges in CTF-DOJO. This yields 252 matched writeups, covering 150 challenges with at least one relevant writeup. During preprocessing, we redact any potential flag values from the writeups and incorporate the cleaned content into the task prompt, as the direct answers may lead to the shortcut learning (Geirhos et al., 2020). Furthermore, two of the authors carefully inspected the matched writeups to confirm that no leaked flags were present and that all writeups corresponded correctly to the CTF challenges. We explicitly instruct the LLM to treat the writeup as a source of inspiration, using its strategies and reasoning implicitly without direct referencing. To ensure the integrity of downstream evaluation, we remove all writeup content from collected trajectories after inference. In addition, to further guarantee that no residual writeup information remains, we randomly sample 20% of the trajectories after this removal step and have two of the authors carefully verify that the agent's reasoning does not reproduce or paraphrase the hint text. This double-check helps confirm that the final data reflect genuinely self-directed problem-solving rather than implicit reuse of the provided hints. *Second, we augment the CTF runtime per agent rollout via* CTF-FORGE *by introducing randomized environment configurations*. These augmentations include varying port numbers, modifying file system paths, injecting non-functional distractor code, and adjusting system-level metadata such as timestamps and installed packages. While preserving the core logic and solvability of each challenge, these perturbations reduce overfitting to static runtime cues and encourage agents to develop more generalizable exploitation strategies. They also help mitigate persistent misconfigurations introduced by LLMs. By resetting the runtime with diverse settings, the environment is more likely to land in a valid configuration that enables flag discovery, even if previous runs failed due to deterministic setup errors. For challenges with dynamic flag generation, we re-seed the container environments at each rollout to ensure unique flag instances per interaction, further enriching training data diversity.

**Data Analysis** We employ two models, Qwen3-Coder (Yang et al., 2025a) and DeepSeek-V3-0324 (Liu et al., 2024), to analyze the composition and characteristics of the raw 1,006 successful trajectories across multiple

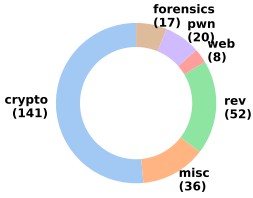

*Figure 2.* Resolution.

runs to better understand the coverage and difficulty distribution within CTF-DOJO. Figure 2 shows the category distribution across solved 274 challenges, where cryptography tasks constitute the largest portion, followed by reverse engineering, and miscellaneous categories. This distribution reflects the typical emphasis in modern CTFs on cryptographic reasoning and binary analysis. We provide more data analysis in Appendix B.

## 3. Training LLMs as Cybersecurity Agents with CTF-DOJO

With CTF-DOJO, we train cybersecurity agents with various base models. Our primary objective is to establish strong baselines and demonstrate the effectiveness of training data derived from execution. We use Pass@$k$ (Chen et al., 2021) as our main evaluation metric. Similar to Pan et al. (2024), we employ a simple policy improvement algorithm: rejection sampling fine-tuning, where we fine-tune the model on trajectories successfully capturing flags inside CTF-DOJO. In addition, we apply sample capping of 2 per solved CTF challenges to avoid bias towards easy tasks, following Pan et al. (2024) and Yang et al. (2025b). We finally collect 486 trajectories from the 274 CTF challenges solved by Qwen3-Coder and DeepSeek-V3-0324 (see Table 5).

### 3.1. Experiment Setup

**Training** We fine-tuned Qwen3 models at three scales: 8B, 14B, and 32B (Yang et al., 2025a). All models undergo supervised fine-tuning on A100 GPUs via NVIDIA NeMo framework (Kuchaiev et al., 2019). Due to computational constraints, we only retain synthesized samples within 32,768 tokens, resulting in 486 trajectories. The hyperparameters are consistently set as the global batch size of 16, the learning rate of 5e-6, and the epoch of 2.

**Evaluation Scaffolding** We use ENIGMA+, an enhanced version of the ENIGMA scaffold with several key improvements for large-scale cybersecurity evaluation. ENIGMA+ executes evaluation tasks in parallel, significantly improving efficiency. Following Zhuo et al. (2025), we cap each rollout at 40 interaction turns, replacing ENIGMA's cost-based budget (Yang et al., 2024a) to ensure consistent evaluation across models. We also adopt the *Simple Summarizer* to prevent context overflows from verbose outputs like binary

*Table 3.* Pass@1 performance on benchmark tasks. The improvements of **CTF-DOJO** are absolute in comparison with the Qwen3 model of corresponding sizes. *: We report the original results of Cyber-Zero. †: To compare with the CTF-DOJO performance, we report the results of Qwen3 models that are trained on a subset of 486 trajectories sampled from the Cyber-Zero data.

| Model | Train Size | InterCode-CTF | NYU CTF | Cybench | Average |
|---|---|---|---|---|---|
| *Proprietary Models* | | | | | |
| Claude-3.7-Sonnet (Anthropic, 2025a) | - | 86.8 | 18.2 | 30.0 | 39.0 |
| Claude-3.5-Sonnet (Anthropic, 2024) | - | 85.7 | 16.7 | 25.0 | 37.2 |
| Gemini-2.5-Flash (Comanici et al., 2025) | - | 81.3 | 14.1 | 17.5 | 33.4 |
| *Open Weight Models* | | | | | |
| DeepSeek-V3-0324 (Liu et al., 2024) | - | 82.5 | 6.2 | 27.5 | 30.3 |
| Kimi-K2 (Team et al., 2025) | - | 72.5 | 4.7 | 15.0 | 25.1 |
| Qwen3-Coder (Yang et al., 2025a) | - | 70.3 | 5.7 | 10.0 | 24.5 |
| Qwen2.5-Coder-7B-Instruct (Hui et al., 2024) | - | 34.1 | 2.0 | 0.0 | 10.8 |
| Qwen2.5-Coder-14B-Instruct (Hui et al., 2024) | - | 44.0 | 3.1 | 5.0 | 14.9 |
| Qwen2.5-Coder-32B-Instruct (Hui et al., 2024) | - | 68.1 | 4.7 | 10.0 | 23.2 |
| Qwen3-8B (Yang et al., 2025a) | - | 46.2 | 0.5 | 5.0 | 14.2 |
| Qwen3-14B (Yang et al., 2025a) | - | 55.0 | 2.6 | 12.5 | 18.6 |
| Qwen3-32B (Yang et al., 2025a) | - | 60.0 | 4.7 | 5.0 | 20.3 |
| Cyber-Zero-8B* (Zhuo et al., 2025) | 9,464 | 64.8 | 6.3 | 10.0 | 23.2 |
| Cyber-Zero-14B* (Zhuo et al., 2025) | 9,464 | 73.6 | 9.9 | 20.0 | 29.1 |
| Cyber-Zero-32B* (Zhuo et al., 2025) | 9,464 | 82.4 | 13.5 | 17.5 | 33.4 |
| Cyber-Zero-8B† (Zhuo et al., 2025) | 486 | 50.5 | 1.0 | 5.0 | 15.4 |
| Cyber-Zero-14B† (Zhuo et al., 2025) | 486 | 57.1 | 3.6 | 12.5 | 19.8 |
| Cyber-Zero-32B† (Zhuo et al., 2025) | 486 | 78.0 | 5.7 | 10.0 | 26.6 |
| CTF-DOJO-8B (Ours) | 486 | 53.8 (7.3% ↑) | 4.2 (3.4% ↑) | 10.0 (5.0% ↑) | 18.9 (4.7% ↑) |
| CTF-DOJO-14B (Ours) | 486 | 71.4 (16.4% ↑) | 5.7 (3.1% ↑) | 17.5 (5.0% ↑) | 25.7 (7.1% ↑) |
| CTF-DOJO-32B (Ours) | 486 | 83.5 (23.5% ↑) | 10.4 (5.7% ↑) | 17.5 (12.5% ↑) | 31.9 (11.6% ↑) |

decompilation.

**Test Benchmarks** We evaluate agents on three established CTF benchmarks detailed in Table 2: InterCode-CTF benchmark comprises 100 CTF challenges collected from picoCTF, an online educational platform for high-school rated CTF challenges. NYU CTF Benchmark contains 200 CTF challenges from CSAW competitions (2017-2023), representing university-level difficulty. Cybench benchmark includes 40 CTF challenges collected from four distinct professional competitions: HackTheBox, Sekai CTF, Glacier and HKCert (2022-2024). These benchmarks collectively span six challenge categories: Cryptography, Forensics, Binary exploitation, Reverse-Engineering, Miscellaneous, and Web. For evaluation, we deploy each LLM within the agent scaffold with access to the Linux Bash terminal.

### 3.2. Result Analysis

We evaluate all LLMs with the Pass@1 metric, where we sample three rollouts per task and validate whether the model captures the correct flag. Following (Zhuo et al., 2025), all the evaluations are under the greedy decoding setting (the temperature of 0.0 and top-p of 0.95), with the maximum agent-environment paired turn as 40. Table 3

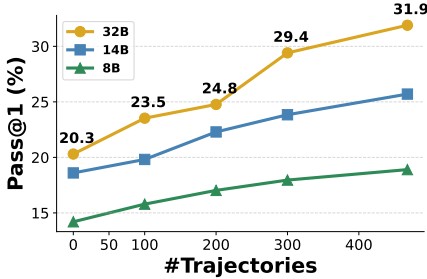

*Figure 3.* Effect of data scaling. Models across sizes benefit from increased number of training trajectories.

presents performance comparisons between zero-shot and fine-tuned models across all benchmarks.

**CTF-DOJO training enables efficient vulnerability exploitation.** Our results show that CTF-DOJO-fine-tuned models achieve performance comparable to Cyber-Zero while using 94.9% fewer training trajectories (486 vs. 9,464). When Cyber-Zero is trained on the same reduced budget of 486 trajectories, CTF-DOJO consistently outperforms it across model scales, indicating substantially higher data efficiency. For example, CTF-DOJO-32B attains an average Pass@1 of 31.9%, surpassing Cyber-Zero-32B trained

on 486 trajectories (26.6%) and approaching the full-data Cyber-Zero-32B result (33.4%). Similar gains are observed for the 14B (25.7% vs. 19.8%) and 8B (18.9% vs. 15.4%) settings. These results demonstrate that competitive performance can be achieved from a compact set of successful CTF trajectories. Notably, CTF-DOJO-trained models also begin to rival frontier systems such as Claude-3.5-Sonnet (37.2%), underscoring the practical feasibility of training capable cybersecurity agents at modest cost.

**Scaling training data improves the performance linearly.**
Figure 3 shows the impact of increasing training trajectories on Pass@1 performance across different model sizes. All model variants (8B, 14B, 32B) demonstrate clear and consistent performance gains as training trajectories increase. Notably, the 32B model improves from 22.0% to 31.9% Pass@1 from 0 to 486 trajectories, demonstrating nearly linear performance scaling with data. This trend confirms that even modestly sized datasets can substantially enhance capability in cybersecurity tasks. Larger models not only start from higher baselines but also benefit more from additional supervision, highlighting the synergistic effect of scale and verified data in training paradigm.

## 4. Ablations on CTF-DOJO Data Collection

To better understand how we can collect more training data, we conduct ablation studies across three axes: external writeups as inference-time hints, runtime augmentation during data collection. These experiments reveal the impact of key design choices and identify practical strategies for enhancing agent performance in cybersecurity environments. We also explore the effectiveness of teacher model diversity in Appendix C. We note that our ablations are based on the assumption of the scaling law (Hoffmann et al., 2022), where models trained on more diverse data tend to achieve better performance.

### 4.1. Writeups as Hints

**Setup**  To assess the value of incorporating external CTF writeups during data collection, we conduct a controlled ablation on CTF-DOJO challenges. We compare two settings: (1) No-Hint (-), where models receive only the original challenge description, and (2) With-Hint (+), where one redacted matched writeups is randomly chosen to prepend to the prompt as a non-referential hint for the corresponding challenge. All other settings remain constant with the main experiments.

**Analysis**  As shown in Table 4, writeup-based hints consistently improve the number of solved tasks across all models and challenge categories. On average, the number of solved challenges increases by 7.4%, from 168 (No-Hint) to

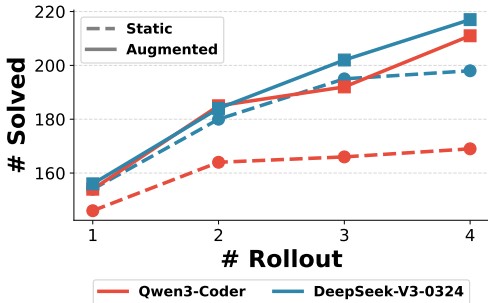

*Figure 4.* Effect of runtime augmentation.

217 (With-Hint), underscoring the utility of public writeups for improving the yield rate of training trajectories. This effect is particularly pronounced in the Crypto, Reverse Engineering, and Miscellaneous categories where solution strategies often rely on reusable heuristics or canonical exploration workflows. This finding suggests that writeups can serve as a rich reservoir of domain-specific knowledge, allowing models to bootstrap strategic reasoning and explore more promising solution paths. We believe the effectiveness of inference-time hints can generalize to various agent tasks like solving GitHub issues (Jimenez et al., 2024), where more diverse data can be distilled from LLMs to train stronger agentic models

### 4.2. Augmenting CTF Runtimes

**Setup**  To evaluate the effect of runtime augmentation on agent performance, we compare two settings for environment construction: (1) Static, where each CTF instance uses fixed runtime parameters, and (2) Augmented, where we introduce perturbations such as randomized port numbers, file path shuffling, distractor code injection, and dynamic flag regeneration. We run both Qwen3-Coder and DeepSeek-V3-0324 across 1 to 4 agent rollouts and count the number of unique CTF challenges successfully solved at least once under each setting. We keep all rollout and decoding hyperparameters identical across both variants to isolate the impact of augmentation.

**Analysis**  Figure 4 shows that augmented environments consistently yield more solved tasks across all rollout counts and both models. For example, Qwen3-Coder solves 211 challenges under augmentation at rollout 4, a relative improvement of 24.9% compared to only 169 under static runtimes. Similarly, DeepSeek-V3-0324 improves from 156 to 217 solved tasks with augmentation at rollout 4. The performance gap widens with more rollouts, suggesting that augmentation amplifies agent exploration and generalization as more interactions are permitted. These results confirm that runtime diversity prevents brittle overfitting to environment artifacts and encourages the development of more

*Table 4.* Solved rate (%) on CTF-DOJO tasks across categories, using ENIGMA+. "–" indicates baseline without writeup hints; "+" includes writeups in the prompt.

| Models | # Crypto | | # Forensics | | # Pwn | | # Rev | | # Web | | # Misc | | # Total | |
|---|---|---|---|---|---|---|---|---|---|---|---|---|---|---|
| | – | + | – | + | – | + | – | + | – | + | – | + | – | + |
| *Proprietary Models* | | | | | | | | | | | | | | |
| Claude-3.7-Sonnet | 41.2 | **50.9** | 42.1 | **50.0** | 14.7 | **20.9** | 41.5 | **49.6** | 61.9 | **76.2** | 47.1 | **69.4** | 36.2 | **46.4** |
| Claude-3.5-Sonnet | 39.9 | **43.9** | 39.5 | **47.4** | 8.0 | **13.5** | 39.8 | **41.5** | 47.6 | **57.1** | 45.9 | **68.2** | 33.0 | **39.7** |
| *Open Weight Models* | | | | | | | | | | | | | | |
| DeepSeek-V3-0324 | 37.1 | **41.0** | 41.0 | **43.6** | 12.0 | **13.5** | 34.1 | **36.6** | 33.3 | **52.4** | 36.5 | **41.2** | 30.4 | **33.9** |
| Qwen3-Coder | 31.4 | **42.8** | 35.9 | **38.5** | 7.9 | **9.1** | 26.8 | **39.8** | 23.8 | **28.6** | 24.7 | **37.6** | 23.9 | **32.5** |
| Qwen3-32B | 21.9 | **29.4** | 7.9 | **18.4** | 1.8 | **6.7** | 22.8 | **28.5** | 9.5 | **23.5** | 31.8 | **41.2** | 17.2 | **24.3** |
| Qwen3-14B | 14.0 | **25.9** | 5.3 | **10.5** | 1.8 | **4.9** | 20.3 | **25.2** | 9.5 | **14.3** | 24.7 | **40.0** | 12.9 | **21.1** |

robust, transferable strategies for flag capture.

## 5. Related Work

**LLM Agents for Offensive Cybersecurity** LLM agents are increasingly being applied to offensive cybersecurity, particularly in solving CTF challenges within dockerized environments (Yang et al., 2023; Shao et al., 2024; Zhang et al., 2025b; Mayoral-Vilches et al., 2025). These systems often build on Kali Linux due to its extensive suite of pre-installed security tools, serving as foundations for broader applications such as penetration testing, vulnerability exploitation, and cyberattack automation (Charan et al., 2023; Deng et al., 2024; Fang et al., 2024). To evaluate the risks and offensive potential of such systems, benchmarks like CyberSecEval (Bhatt et al., 2023; Wan et al., 2024) have been proposed, while others assess the "dangerous capabilities" of LLMs in tasks like CTFs and red-teaming (Phuong et al., 2024; Guo et al., 2024), though these models still show limited performance on more complex tasks. Recent efforts have advanced agent design. Project Naptime (Glazunov & Brand, 2024) and Big Sleep (Allamanis et al., 2024) demonstrated agents capable of discovering new SQLite vulnerabilities using integrated tools like debuggers and browsers. EnIGMA (Abramovich et al., 2025) further raises the bar by combining cybersecurity-specific tools and interactive environments tailored for LLMs, achieving state-of-the-art results. Recently, Zhuo et al. (2025) introduced Cyber-Zero, achieving the best performance among open-source LLMs. Unlike prior methods that primarily depend on inference-time scaffolds or unverified training data, we introduce a runtime environment that efficiently enhances model performance via execution.

**Benchmarking Models' Cybersecurity Capabilities** Several benchmarks have been proposed to evaluate LLMs on cybersecurity tasks. Multiple-choice datasets (Li et al., 2024; Tihanyi et al., 2024; Liu, 2023) offer limited in-sight, as their results are often highly sensitive to prompt phrasing (Qi et al., 2024; Łucki et al., 2024) and lack alignment with real-world operational contexts. AutoAdvExBench (Carlini et al., 2025) assesses LLMs' ability to autonomously break image-based adversarial defenses, while CyberSecEval (Bhatt et al., 2023) focuses on single-turn code exploitation, capturing only a narrow slice of the interactive, multi-step nature of real-world attacks. In contrast, agent-based frameworks with integrated tool usage offer more realistic evaluations. As a result, Capture-the-Flag (CTF) challenges have become a popular proxy for measuring security capabilities. Recent systems (Abramovich et al., 2025; Mayoral-Vilches et al., 2025) further enhance realism by combining interactive environments with structured, chain-of-exploitation evaluations.

## 6. Conclusion and Future Work

**Conclusion** We introduce CTF-DOJO, the first large-scale execution environment for training cybersecurity LLM agents, addressing the lack of runtime support in this domain. Powered by our automated pipeline CTF-FORGE, CTF-DOJO converts public CTF artifacts into ready-to-use Docker containers in minutes, enabling scalable and reproducible trajectory collection. Trained on only 486 high-quality agent trajectories, our open-weight LLMs outperform strong baselines by up to 11.6% on three major CTF benchmarks. Our 32B model achieves outstanding results among open models, approaching Claude-3.5-Sonnet and DeepSeek-V3-0324. These findings highlight the importance of hinted training, runtime augmentations, and diverse agent behaviors, positioning CTF-DOJO as a scalable and accessible foundation for LLM-based security systems.

**Future Work** This work highlights several directions for future research. We envision a live CTF benchmark that continuously evaluates models on challenges from active competitions. Using CTF-FORGE to dynamically recon-

struct and containerize environments enables scalable, real-time benchmarking while mitigating contamination risks. The static and limited scale of CTF-DOJO motivates exploring reinforcement learning to support more generalizable strategies through partial or flag-based rewards. Finally, although we focus on the `pwn.college` CTF Archive, CTF-FORGE is source-agnostic. Extending to more diverse CTF repositories will require stronger environment-configuration strategies, potentially combined with agentic methods for dependency inference and build validation.

## Impact Statement

This work introduces CTF-DOJO and CTF-FORGE, enabling scalable, execution-verified training data for cybersecurity agents and improving the evaluation and development of models that can discover and validate software flaws in realistic CTF-style environments. The primary positive impact is advancing defensive security research by lowering the barrier to reproducible experimentation, enabling more systematic study of agent reliability, and helping identify brittle failure modes before deployment. At the same time, this work is inherently dual use, where the same capabilities that support vulnerability discovery for defense can be misused to automate exploitation or accelerate offensive operations. We mitigate this risk through secure container isolation, careful dataset construction (including redaction of flags and removal of writeup text from released trajectories), and by positioning CTF-DOJO as a research and training environment rather than a deployment tool. Nonetheless, misuse remains possible, and we encourage responsible release practices, auditing of downstream applications, and alignment and access controls when integrating such agents into real-world security workflows.

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

# A. Statistics

We provide a summary of the important statistics mentioned in the paper.

*Table 5.* Summary of data statistics.

| Item Description | Count |
|---|---|
| CTF-DOJO *Challenges* | |
| Number of available CTF challenges | 658 |
| Number of challenges with stable and reproducible environments, as confirmed by the original authors | 650 |
| *Writeups for CTF Challenges* | |
| Total number of writeups collected from the CTFtime website | 8,361 |
| Writeups successfully matched to CTF-DOJO challenges using competition and task metadata | 252 |
| CTF-DOJO challenges for which at least one corresponding writeup is available | 150 |
| *Successful Agent Samples* | |
| Raw agent trajectories collected before cleaning or filtering | 1,006 |
| Unique trajectories remaining after removing duplicates and limiting the maximum number per challenge | 486 |
| CTF-DOJO challenges that include at least one valid and successful trajectory | 274 |

## B. Data Analysis

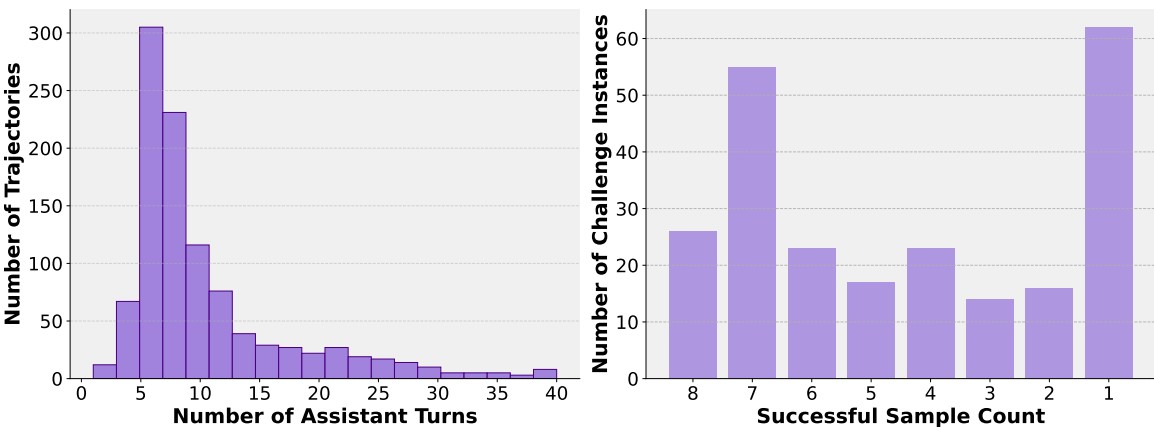

*Figure 5.* Number of turns in each successful trajectory (left) and number of successful trajectories for each challenge instance (right).

Figure 5 presents two key statistics of the collected data. The left panel visualizes the number of assistant turns per trajectory. The majority of trajectories fall between 5 to 15 turns, with a heavy tail extending to 40 turns. This skew indicates that while many tasks can be solved efficiently, a substantial portion demands prolonged, iterative explorations, highlighting the complex nature of real-world CTF problems. The right panel plots the number of successful trajectories obtained for each challenge, revealing that many challenges are solved only once within the total 12 rollouts, indicating that successful trajectories for certain instances are difficult to collect.

## C. More Ablation Studies

**Setup** To assess the benefit of using multiple teacher models during trajectory collection, we compare the individual and combined contributions of Qwen3-Coder and DeepSeek-V3-0324. We first analyze how many unique challenges each model solves and their category-level overlaps. Then, we fine-tune Qwen3 models of sizes 8B, 14B, and 32B on three trajectory subsets: (1) Qwen3-Coder only, (2) DeepSeek-V3-0324 only, and (3) both combined. We report average Pass@1 across benchmarks to evaluate downstream agent performance. Decoding parameters and training setup match those in our main experiments.

| Category | Qwen | Both | DeepSeek |
|---|---|---|---|
| **Crypto** | 31 | **84** | 26 |
| **Forensics** | 1 | **13** | 3 |
| **Pwn** | 2 | **15** | 3 |
| **Rev** | 6 | **37** | 9 |
| **Web** | 0 | **6** | 2 |
| **Misc** | 4 | **26** | 6 |

*Table 6.* Solved challenge counts.

**Analysis** In Table 6, Qwen3-Coder and DeepSeek-V3-0324 demonstrate complementary strengths. For example, in Crypto tasks, the models share 84 solves, but Qwen3-Coder uniquely solves 31 while DeepSeek-V3-0324 adds another 26. Similar patterns emerge across other categories, with notable non-overlapping contributions in Reverse Engineering, Misc, and Forensics. Combining both models increases total coverage to 274 unique challenges, exceeding either model alone. This diversity translates into measurable downstream gains.

*Table 7.* Pass@1 performance when varying teacher models.

| Teacher Model | 8B | 14B | 32B |
|---|---|---|---|
| Qwen3-Coder | 17.3 | 23.8 | 29.4 |
| DeepSeek-V3-0324 | 17.6 | 24.8 | 31.3 |
| Combined | **18.9** | **25.7** | **31.9** |

Table 7 reveals that training on combined trajectories improves Pass@1 performance across all model sizes. For example, the 32B model trained on combined data achieves 31.9%, outperforming both the Qwen3-Coder-only (29.4%) and DeepSeek-only (31.3%) variants. Similarly, the 8B and 14B models also benefit from the combined setting. These results confirm that

teacher diversity enriches training data and yields more capable cybersecurity agents.

## D. More Related Work

**Training LLM Agents to Code** Previous training paradigms for software engineering have largely emphasized general-purpose coding capabilities (Li et al., 2023; Lozhkov et al., 2024; Muennighoff et al., 2024; Zhuo et al., 2024; Wei et al., 2024). While scaffolded approaches using proprietary models achieve strong results on real-world software engineering (SE) tasks, open-source models continue to lag behind, prompting a shift toward domain-specific training strategies. Several recent efforts exemplify this trend. Lingma SWE-GPT (Ma et al., 2024) introduces 7B and 72B models trained with a process-oriented development methodology. SWE-Gym (Pan et al., 2024) offers the first open training environment for SE agents, yielding notable gains on SWE-bench (Jimenez et al., 2024). More recent work includes SWE-smith (Yang et al., 2025b), which automatically scales training data for SE, and SWE-RL (Wei et al., 2025), which applies reinforcement learning (Grattafiori et al., 2024) to repair programs with reasoning. While these methods advance software engineering capabilities via execution-based environments, they do not address the distinct demands of cybersecurity (Zhuo et al., 2025). Our work fills this gap by introducing a large-scale execution environment specifically tailored for security tasks, where traditional code-centric training fails to transfer effectively.

## E. Scaffolding Interface

We simulate the ENIGMA Scaffold interface in CTF-DOJO, and provide specialized tools inside Table 8 from the original ENIGMA paper (Abramovich et al., 2025). While we provide the interface to the models for data generation, there is no guarantees that they will utilize the customized tools regularly.

*Table 8.* In additional to the standard Linux Bash commands and the SWE-agent specialized tools, we provide ENIGMA with tools in the offensive cybersecurity domain, including binary decompilation and disassemble, and interactive agent tools for debugging and connecting to remote servers. Required arguments are enclosed in <> and optional arguments are in []. The last column shows the documentation presented to the LLMs.

| Category | Command | Documentation |
|---|---|---|
| *Static analysis* | **decompile** `<binary_path>` `[-function_name <function_name>]` | Decompile a binary and prints the decompilation of a given function name, or main by default. |
| | **disassemble** `<binary_path>` `[-function_name <function_name>]` | Disassembles a binary and prints the disassembly of a given function name, or main by default. |
| *Dynamic analysis* | **debug_start** `<binary>` `[<args>]` | Starts a debug session with the given binary and optional command-line arguments for the binary. |
| | **debug_add_breakpoint** `<breakpoint>` | Adds a breakpoint in the debug session. |
| | **debug_continue** **debug_step** `[<number>]` **debug_exec** `<command>` **debug_stop** | Continues the program execution in the debug session. Steps number of instructions (default is 1) in the debug session. Executes arbitrary `gdb` command in debug session. Stops the current debug session. |
| *Network* | **connect_start** `<server_address>` `<port>` | Starts a new interactive connection to the server address and port. |
| | **connect_sendline** `[<line>]` | Sends a line with Unicode/hexadecimal values to the connection. Sending hexadecimal bytes should be done using \x<hh> where hh stands for the specific byte you want to send. |
| | **connect_exec** `<command>` **connect_stop** | Executes arbitrary connect command in connect session. Stops the current connect session. |
| *Task* | **submit** `'<flag>'` | Submits your current flag and terminates the session upon correct response, you should be aware to properly escape the flag as this is a bash command and to put your flag in single quotes. |
| | **exit_forfeit** | Give up on the current challenge and terminate the session. |

# F. Prompt Design of CTF-FORGE

## F.1. Dockerfile Generation

```
Based on the following CTF challenge information, generate a Dockerfile that will properly set up a server environment for CTF ⟵
    players to interact with.

# Challenge Name:
{task_name}

# Challenge Category:
{category}

# Available Files:
{available_files}

# File Analysis:
{file_analysis}

# Challenge Description:
{description}

# REHOST.md Content:
{rehost_content}

# .init File Content:
{init_content}

**IMPORTANT**: If .init file content is provided above, this contains setup instructions or configuration specific to this challenge⟵
    . The .init file may contain:
- Environment setup commands that should be executed during Docker build
- Configuration parameters or paths that should be used in the Dockerfile
- Special instructions for handling this particular challenge
- Library or dependency information
- Runtime configuration that affects how the challenge should be containerized

Use the .init content to inform your Dockerfile generation - execute any setup commands it specifies, copy any files it references, ⟵
    and follow any special instructions it provides.

{flag_instruction}

# CATEGORY-SPECIFIC GUIDELINES:
{category_guidelines}

# GENERAL DOCKER BEST PRACTICES:
1. Use ubuntu:20.04 as base image unless challenge specifically requires different environment
2. Install additional packages needed for the specific challenge (if any beyond the comprehensive set)
3. Create non-root user for running services when possible, but DO NOT use `useradd`
4. Use specific package versions when critical for challenge functionality
5. Properly set file permissions (especially for executables and sensitive files)
6. Use WORKDIR to organize files properly
7. Expose only the necessary port for the challenge
8. Use specific CMD/ENTRYPOINT that matches the challenge type
9. You MUST create a `run.sh` wrapper script using printf or echo commands (DO NOT use heredoc syntax like << 'EOF')

# CRITICAL SCRIPT CREATION SYNTAX:
- CORRECT: Use printf to create shell scripts in Dockerfiles:
  ```
  RUN printf '#!/bin/bash\\ncd /challenge\\nexec stdbuf -i0 -o0 -e0 ./your_executable\\n' > /challenge/run.sh
  ```
- CORRECT: Use echo with \\n for newlines:
  ```
  RUN echo -e '#!/bin/bash\\ncd /challenge\\nexec stdbuf -i0 -o0 -e0 ./your_executable' > /challenge/run.sh
  ```
- WRONG: Never use heredoc syntax in Dockerfiles (causes parsing errors):
  ```
  RUN cat > /challenge/run.sh << 'EOF'  # This will FAIL
  #!/bin/bash
  cd /challenge
  exec stdbuf -i0 -o0 -e0 ./your_executable
  EOF
  ```

# CRITICAL EXECUTABLE/SCRIPT WRAPPING REQUIREMENTS:
{architecture_specific_wrapper}

# SECURITY CONSIDERATIONS:
- DO NOT copy flag.sha256 or flagcheck files to the Docker image
- DO NOT copy any files not in the available files list
- Set appropriate file permissions for challenge files
- Use process isolation when running network services

# SHEBANG HANDLING:
- If any files have problematic shebangs (like #!/opt/pwn.college/python), they will be automatically fixed
- You do not need to manually handle shebang issues - the system will detect and fix them
- Focus on proper file copying and permissions

# CTF-SPECIFIC REQUIREMENTS:
- The server MUST be accessible to CTF players over the network
- The flag should NEVER be directly accessed by the CTF players, and MUST always be stored in /flag with the permissions 444
- Choose appropriate port based on service type (1337 for general, 80/8080 for web, etc.)
```

```
- Ensure the challenge service starts automatically and runs continuously
- Handle connection multiplexing for multiple players if needed
- Programs must respond to user input immediately without buffering delays (achieved through stdbuf)

# CRITICAL CTF BINARY BEHAVIOR UNDERSTANDING:
- CTF challenge binaries (especially pwn challenges) often exhibit specific behavior patterns:
  * When run directly from command line, they may exit immediately without output (THIS IS NORMAL)
  * They are designed to work through network services (socat) that provide stdin/stdout redirection
  * The binary may wait for specific input patterns or network connections to respond
  * Some binaries are designed to read from stdin and write to stdout in an interactive manner
- Do NOT assume a binary is broken if it runs without output when executed directly
- The key is to properly wrap the binary with socat for network access
- Test the service through network connection (nc localhost PORT) rather than direct execution

# LIBRARY DEPENDENCY HANDLING:
- Pay special attention to shared library dependencies (check with ldd if needed conceptually)
- For 32-bit binaries on 64-bit systems, ensure 32-bit libraries are installed
- If a binary requires specific libraries (e.g., libpam.so.0), install the appropriate packages:
  * For libpam: install libpam0g:i386 for 32-bit or libpam0g for 64-bit
  * Use library path environment variables or LD_LIBRARY_PATH if needed
  * Consider using the system's dynamic linker directly for better compatibility

Generate a complete, production-ready Dockerfile. Respond with ONLY the Dockerfile content, no explanations.

IMPORTANT VALIDATION CHECKLIST:
Base image specified (prefer ubuntu:20.04)
NOTE: Comprehensive package installation will be automatically added
Additional required packages installed (if needed beyond the comprehensive set)
Challenge files copied correctly
run.sh wrapper script created with stdbuf for proper interaction
Appropriate port exposed
Service command specified in CMD/ENTRYPOINT using socat with the run.sh wrapper
File permissions set correctly for both executables and run.sh
No sensitive files copied
Service will accept network connections and respond immediately to user input
CRITICAL: Scripts created using printf/echo commands, NOT heredoc syntax (<<)

#  CRITICAL DOCKERFILE SYNTAX WARNING:
- NEVER use heredoc syntax like "RUN cat > file << 'EOF'" in Dockerfiles
- This causes Docker parsing errors and build failures
- ALWAYS use printf or echo commands instead
- Example: RUN printf '#!/bin/bash\\ncd /challenge\\nexec ./binary\\n' > /challenge/run.sh

# PYTHON NETWORK SERVICES:
- If the file analysis indicates a Python script is a network server listening on a specific internal port (e.g., detected as ↩
    listening on port XXXX):
- The service MUST be run in the background (e.g., `python3 /challenge/server.py &`).
- You MUST use `socat` to proxy connections from the public EXPOSED port (e.g., 1337) to the script's detected internal port.
- **CORRECT WAY** to create `run.sh` for a Python server on its detected internal port, exposed on 1337:
  ```
  RUN printf '#!/bin/sh\\ncd /challenge\\n# Start the server in the background\\npython3 /challenge/server.py &\\n# Wait a moment ↩
      for the server to start\\nsleep 1\\n# Use socat to forward connections from the public port to the internal port\\nexec socat ↩
      TCP-LISTEN:1337,reuseaddr,fork TCP:localhost:XXXX\\n' > /challenge/run.sh && chmod +x /challenge/run.sh
  ```
- The `CMD` in the Dockerfile should then be `CMD ["/challenge/run.sh"]`.
- DO NOT use `socat` with `EXEC` for these types of services, as it launches a new process for every connection.
```

## F.2. Docker-Compose Generation

```
Based on the following CTF challenge information and generated Dockerfile, create a docker-compose.yml file with an appropriate ↩
    ctfnet alias.

# Challenge Name:
{task_name}

# CTF Name:
{ctf_name}

# Available Files:
{available_files}

# Challenge Description:
{description}

Generated Dockerfile:
{dockerfile_content}

Requirements:
1. Use "build: ." to build from the local Dockerfile
2. Extract the exposed port from the Dockerfile and map it appropriately
3. Connect to the "ctfnet" external network
4. Generate a meaningful, DNS-compliant alias for this specific challenge
5. The alias should be memorable and related to the challenge name/theme
6. Use format like: challengename.ctf.io or similar creative naming
7. Avoid generic names like "web.chal.custom.io"
8. Use the challenge information to create a relevant alias

Example good aliases:
- showdown.csaw.io
```

```
- cryptochallenge.picoctf.io
- webshell.defcon.io
- bufferoverflow.ctf.io

Generate a complete docker-compose.yml. Respond with ONLY the docker-compose.yml content, no explanations.'
```

## F.3. Challenge.json Generation

```
Generate a challenge.json file for this CTF challenge following the exact schema.

Challenge Information:
# Name:
{task_name}

# Category:
{category}

# Files:
{task_files}

# CTF Challenge RAW Description:
{description}

# REHOST.md Content:
{rehost_content}

{docker_compose_section}

REQUIRED JSON SCHEMA:
{{
    "name": "challenge name",
    "description": "human readable and self-contained challenge description for the CTF players, not for the CTF rehosters. flag ←┘
      checker is NOT available and you should NOT mention it in the description",
    "category": "will be set automatically from module.yml mapping",
    "files": ["list of files to upload for this challenge"],
    "box": "optional URL for external server challenge",
    "internal_port": "optional port for external server challenge",
    "compose": true if hosting the ctf challenge requires a server, false otherwise,
    "flag": "flag for the challenge"
}}

RULES:
1. Use the exact challenge name: "{task_name}"
2. Use the exact files list: {task_files}
3. DO NOT generate the category field - it will be set automatically from module.yml mapping
4. Always set "box": "" and "internal_port": ""
5. If server_needed is False: do NOT include "server_description" field
6. The flag should be a valid flag for the challenge
7. You MUST NOT mention the flagCheck in the Challenge Description, as we will handle the validation in other way.

OUTPUT: Return ONLY valid JSON matching the schema above. No explanations, no markdown, no extra text.
```

# G. Finding Bugs in CTF-DOJO

Through our experiments on CTF-DOJO, we identify 4 bugs inside the current CTF-Archive maintain by pwn.college. Specifically, we observe a consistent behavior pattern across these problematic challenges, where the LLMs make several attempts to submit the same flag mismatched with the one encoded inside the original repository.

## G.1. ECTF 2014 — Lowkey

> **CTF Challenge**
>
> Hey! I just met you! And this is craazy! But here's a product of prime numbers! Try me maybe! 2449 P.S. My mother told me that the 7th key fits in the lock ;)
> Hint: Hint is implied in the question.

**Solution** To solve the `lowkey` cryptography challenge, the LLM began by extracting the contents of the provided archive `low_key.tar.gz`, which revealed a file named `whatami.txt` containing a sequence of integers. The challenge description hinted at a "product of prime numbers" (2449) and a "7th key," suggesting an RSA encryption scheme with a public exponent $e = 7$. The LLM factored 2449 into its prime components, $p = 31$ and $q = 79$, yielding $n = pq = 2449$ and Euler's totient $\phi(n) = (p-1)(q-1) = 2340$. Using the extended Euclidean algorithm, the LLM computed the modular inverse of 7 modulo 2340 to obtain the private key exponent $d = 1003$. With this, the encrypted integers from

whatami.txt were decrypted using the formula $m = c^d \bmod n$, and the resulting values were converted to ASCII characters. This will reveal the flag in the format of flag{...}.

**Verification**   The model attempts multiple times to submit the same flag but got rejected, as it did not align with the flag hashed by the pwn.college team. To verify the validity of captured flag, we have done some research on the possible ECTF2014 writeups online but could not find any of them. However, when searching for the flag content captured by the model, we notice there is a blog in Chinese that describes the similar CTF challenge and confirms the flag correctness.

### G.2. ångstromCTF 2019 — Blank Paper

> **CTF Challenge**
>
> Someone scrubbed defund's paper too hard, and a few of the bytes fell off.

**Solution**   To solve the blankpaper challenge, the LLM identified that the PDF file blank_paper.pdf was corrupted due to missing header bytes. By inspecting the file with a hex viewer, it discovered that the expected %PDF signature had been replaced with null bytes. It then created a script to restore the missing header by replacing the first four bytes with %PDF. After regenerating the corrected PDF as fixed_paper.pdf, the model used pdftotext to extract its contents. Within the text, it found the flag in the format of actf{...}.

**Verification**   As the flag format is not specified in the challenge description, the model has tried with the flag wrapper of flag{...} and actf{...}. Both of them fails the flagCheck and .flag.sha256. However, the submitted context inside the flag remains unchanged. We validate the flag using a writeup shown on CTFtime.

### G.3. HSCTF 2019 — Hidden Flag

> **CTF Challenge**
>
> This image seems wrong.....did Keith lose the key again?

**Solution**   To solve the hiddenflag miscellaneous CTF challenge, the LLM was given a file named chall.png, which, although named as a PNG image, was identified by the file command as generic data. Upon inspecting the file using strings, the clue key is invisible was discovered. This led to the hypothesis that the file was XOR-encrypted using the key invisible. A Python script was created to XOR-decrypt the file byte-by-byte using this key. The output, saved as decrypted.png, was confirmed to be a valid PNG image. Optical character recognition (OCR) was then performed using Tesseract, which successfully extracted the flag embedded in the image.

**Verification**   The model made the same flag submission attempts for several times but all of them failed. We find a writeup on the personal website that describes the similar solution and the flag value same as what the model captures.

### G.4. Access Denied CTF 2022 — Binary

> **CTF Challenge**
>
> Finally, you are in the binary stage.

**Solution**   To solve the hiddenflag CTF challenge, the LLM was provided with a file named chall.png, which was not recognized as a valid PNG file. Upon running strings on the file, we found the phrase key is invisible, suggesting XOR encryption with the key invisible. A Python script was used to XOR each byte of the file with the repeating key, producing a valid image saved as decrypted.png. After confirming the decrypted file was a PNG, we ran OCR using Tesseract to extract any hidden text. The extracted text revealed the flag in the format of hsctf{...}.

**Verification** The flag submitted by the model does not match with the officially provided hash in the repository. We confirm the correctness of the submission via a writeup written in the personal blog.

