# OpenReview forum: "Training Language Model Agents to Find Vulnerabilities with CTF-Dojo"
_ICML.cc/2026/Conference — ICML 2026 regular_

### Official Review · Reviewer_Bwnz · 2026-02-25

**Soundness:** 3
**Presentation:** 3
**Significance:** 2
**Originality:** 2
**Overall Recommendation:** 4
**Confidence:** 2

**Summary:**

This paper introduces CTF-DOJO, a large-scale executable training environment for cybersecurity agants. And the authors proposes an automated pipeline: CTF-FORGE that converts the public CTF artifacts into runnable containers. After fine-tuning on LLM backends, the paper shows consistent improvement over pass@1 against baseline methods. At last, the paper discusses data collection strategies.

**Compliance With Llm Reviewing Policy:**

Affirmed.

**Final Justification:**

The concerns have been addressed, so I changed the score accordingly.

**Key Questions For Authors:**

1. How does the author tackle the inherent hallucination issue of using LLM?

2. Can you provide more details on the length of the trajectories? Does the trajectory composition affect the overall LLM context capability?

**Limitations:**

1. Spell out a concrete release plan and safeguards: Specify exactly what will be released (containers, build scripts, metadata, trajectories, fine-tuned weights), what will not be released (e.g., exploit-ready payloads), and whether access will be gated (request-based access, rate limits, click-through terms, institutional affiliation, etc.).

2. Include risk-mitigation experiments or analyses: E.g., evaluate whether the fine-tuned models produce more actionable exploit code on non-CTF targets, or add a red-team style analysis of harmful capability lift; alternatively, provide refusal/policy behavior checks if relevant.

**Strengths And Weaknesses:**

# Strengths

1. The paper provides a executable, verifiable training signal
2. 658 dockerized challenges are meaningfully larger than many prior benchmarks.
3. Beneficial ablations yield a great hint on how to collect more successful trajectories without massive labeling.
4. The collection pipeline includes filtering and verification steps, which improve dataset quality relative to unverified logs.


# Weaknesses
1. The evaluation story would be more convincing with stronger controls for leakage / contamination given reliance on public CTF artifacts and writeups. Even if flags are removed from trajectories, memorized solutions or overlapping challenges can inflate results unless rigorously ruled out.
2. Several experimental choices are under-specified for full rigor: e.g., exact teacher model settings, rollout budgets, sampling strategy, tool/scaffold parity across systems, and how hyperparameters were selected. This makes it harder to assess whether improvements stem from the dataset, scaffold, or tuning choices.

---

> ### Author Rebuttal · Authors · 2026-03-31
>
> Thanks for your valuable review. We address your concerns below:
>
> > W1: The evaluation story would be more convincing with stronger controls for leakage/contamination.
>
> We thank the reviewer for raising this important concern. Our decontamination procedure identifies and matches challenges across three levels of granularity: the specific competition name, the competition year, and the individual challenge name, comparing these identifiers between the pwn.college CTF Archive and all evaluation benchmarks. The pwn.college CTF Archive itself only contains non-duplicated challenges by design, providing an additional guarantee against within-dataset duplication. Based on this matching process, we manually confirmed and removed 3 CTF challenges overlapping with evaluation benchmarks. Furthermore, the Archive aggregates challenges from distinct competition events hosted between 2011 and 2025, where each event is independently organized and typically does not reuse challenges, significantly reducing the risk of near-duplicate contamination. We will add this more detailed description to the final version to make the process more transparent.
>
> > W2: Several experimental choices are under-specified for full rigor.
>
> We have detailed the experimental settings in the Trajectory Collection part under Section 2.4. Specifically, we build on ENIGMA+ [1], deploying Qwen3-Coder and DeepSeek-V3-0324 with a temperature of 0.6, top-p of 0.95, and rollout count of 6. All compared methods in our evaluation use the same ENIGMA+ scaffold to ensure fair comparison. We will consolidate these details more prominently and include a full hyperparameter table in the final version to improve reproducibility and transparency.
>
> > How does the author tackle the inherent hallucination issue of using LLM?
>
> We address hallucination at two levels. At the environment generation level, CTF-Forge employs predefined category-specific templates that constrain generation to structured formats, reducing the risk of invalid configurations, and the automated validation procedure filters out any environments that fail to build or respond correctly. At the trajectory collection level, we rely on execution-verified trajectories, retaining only trajectories where the agent successfully captures the flag. This execution-based verification serves as a strong filter against hallucinated solution steps, as the agent must interact with a real runtime and produce a valid flag to have its trajectory included in training data.
>
> > Can you provide more details on the length of the trajectories? Does the trajectory composition affect the overall LLM context capability?
>
> We provide a thorough analysis of trajectory statistics in Appendix B. As shown in Figure 5, the majority of successful trajectories fall between 5 to 15 assistant turns, with a heavy tail extending to 40 turns, indicating that while many tasks can be solved efficiently, a substantial portion demands prolonged iterative exploration. We set the maximum agent-environment paired turns to 40, following prior work [2] finding that cybersecurity agents tend to solve tasks within 40 steps. Regarding context capability, we only utilize successfully completed trajectories for training, ensuring the model learns from complete and coherent solution paths rather than truncated interactions, avoiding context management issues that might arise from incomplete trajectories.
>
> > Concrete release plan and safeguards.
>
> We have addressed responsible release in our Impact Statement. We mitigate dual-use risks through secure container isolation, careful dataset construction including redaction of flags and removal of writeup text from released trajectories, and by positioning CTF-Dojo as a research and training environment rather than a deployment tool. We will expand this discussion in the final version to include a concrete release plan specifying what will be released, including Docker build scripts, challenge metadata, and execution-verified trajectories, as well as access control mechanisms to ensure materials are used responsibly within the research and educational community.
>
> [1] Zhuo, T. Y., et al. (2025). Cyber-zero: Training cybersecurity agents without runtime. ICLR.
>
> [2] Abramovich, T., et al. EnIGMA: Interactive Tools Substantially Assist LM Agents in Finding Security Vulnerabilities. ICML.

---

> > ### Author Rebuttal · Reviewer_Bwnz · 2026-04-03
> >
> > Thanks for the clarification. I don't have any further questions.

---

> > > ### Author Response · Authors · 2026-04-04
> > >
> > > We sincerely thank Reviewer Bwnz for the thorough and constructive review, and for taking the time to carefully read our rebuttal. We are glad that our responses have fully addressed your concerns.
> > >
> > > We would like to kindly draw the reviewer's attention to the fact that, in light of the resolved concerns, the current score of 3 (Weak Reject) may no longer accurately reflect the state of the paper. Given that all weaknesses have been addressed, including the decontamination procedure, experimental specification, hallucination mitigation, trajectory analysis, and the release plan, we would be grateful if the reviewer could consider whether an upward score adjustment would be appropriate. We believe the clarifications provided in the rebuttal, together with our commitment to incorporating all suggested revisions into the final version, have substantially strengthened the paper's presentation and rigor.
> > >
> > > We deeply appreciate the reviewer's time and constructive engagement throughout this process, and we remain happy to address any further questions or concerns the reviewer may have.

---

### Official Review · Reviewer_Xiuj · 2026-03-13

**Soundness:** 3
**Presentation:** 3
**Significance:** 3
**Originality:** 3
**Overall Recommendation:** 5
**Confidence:** 4

**Summary:**

This paper introduces CTF-DOJO, a large-scale executable environment for training language-model agents on cybersecurity tasks using Capture-The-Flag challenges, and CTF-FORGE, an automated pipeline that transforms public CTF artifacts into reproducible Dockerized runtimes. The core idea is to move beyond non-executable or purely synthetic supervision by collecting execution-verified agent trajectories in realistic environments. The paper reports 658 CTF challenges in total, with 650 stable environments, and constructs a filtered training set of 486 successful trajectories from 274 solved challenges.

**Compliance With Llm Reviewing Policy:**

Affirmed.

**Final Justification:**

The author addressed my concerns.

**Key Questions For Authors:**

**How rigorous is the decontamination procedure with respect to evaluation benchmarks and public writeups?**

Please clarify whether decontamination was done only by exact task identity, or also by near-duplicate challenge families, repeated challenge names, same-source competition overlap, and writeup similarity. A stronger answer here would increase my confidence that gains are not partly due to benchmark-adjacent leakage.

**What is the category and difficulty distribution of the final 486 retained training trajectories?**

Since cryptography seems heavily represented among solved tasks, I would like to know whether the downstream gains are similarly concentrated in some categories. A category-wise downstream evaluation or transfer analysis would help assess generality.

**Limitations:**

yes

**Strengths And Weaknesses:**

Strengths:

1. Timely and important problem setting. The paper addresses a highly relevant gap in current LLM-agent research: the lack of scalable, executable training environments for offensive cybersecurity tasks.

2. Strong systems contribution. CTF-DOJO and CTF-FORGE together constitute a meaningful infrastructure contribution. Building 658 containerized CTF tasks, with reported 98% reliability across repeated runs, is nontrivial and potentially valuable for the community.

3. Clear empirical improvements. The reported fine-tuning results are strong. With only 486 trajectories, the 32B model improves from 20.3 to 31.9 average Pass@1, and also outperforms a budget-matched Cyber-Zero variant. This suggests that execution-verified trajectories are high-value supervision. The data efficiency claim is one of the strongest aspects of the paper.

Weaknesses:

1. Novelty is somewhat mixed between infrastructure and learning method. The strongest contribution is the environment and data pipeline; the actual learning recipe is comparatively simple rejection-sampling fine-tuning on successful trajectories. This is perfectly reasonable, but it means the main algorithmic novelty is limited. The paper’s central novelty is therefore mainly infrastructural rather than methodological.

2. Claims about “pivotal” importance may be a bit too strong for the current evidence. The results convincingly show that execution-verified trajectories are useful and data-efficient. However, the broader claim that such signals are “pivotal” may be stronger than what is strictly established here, especially since comparisons are limited to a specific baseline family and a single downstream task domain.

---

> ### Author Rebuttal · Authors · 2026-03-31
>
> Thanks a lot for your valuable review. We would like to address your concerns as follows:
>
>
> > How rigorous is the decontamination procedure with respect to evaluation benchmarks and public writeups?
>
>
> We thank the reviewer for raising this important concern. We would like to clarify that our decontamination procedure is more rigorous than what may appear from the paper's description. Specifically, we identify and match challenges across three levels of granularity: the specific competition name, the competition year, and the individual challenge name, comparing these identifiers between the pwn.college CTF Archive and the challenges used in all evaluation benchmarks. The pwn.college CTF Archive itself only contains non-duplicated challenges by design, which provides an additional layer of guarantee against within-dataset duplication. Based on this matching process, we manually confirmed and removed 3 CTF challenges that overlap with the evaluation benchmarks. We will add this more detailed description of the decontamination procedure to the final version of the paper to make this process more transparent and rigorous to readers.
>
>
> Regarding near-duplicate challenge families and writeup similarity, we note that the pwn.college CTF Archive aggregates challenges from distinct competition events hosted between 2011 and 2025, where each competition event is independently organized and typically does not reuse challenges across events. This structural property of the archive significantly reduces the risk of near-duplicate contamination beyond what our explicit identifier-based matching already handles. We will make this point clearer in the final version.
>
>
> > What is the category and difficulty distribution of the final 486 retained training trajectories?
>
>
> We thank the reviewer for this question and agree that a clearer presentation of the category distribution would strengthen the paper. As shown in Figure 2 of the paper, the category distribution across the 274 solved challenges is as follows: cryptography constitutes the largest portion with 141 challenges, followed by reverse engineering with 52 challenges, miscellaneous with 36 challenges, forensics with 17 challenges, pwn with 20 challenges, and web with 8 challenges. We acknowledge that cryptography is indeed heavily represented in the solved challenges, which is consistent with the finding in our ablation study that writeup-based hints are particularly effective for cryptography tasks where solution strategies often rely on reusable heuristics.
>
>
> Regarding whether the downstream gains are similarly concentrated in certain categories, we note that our evaluation benchmarks including InterCode-CTF, NYU CTF Bench, and Cybench contain diverse challenge categories, and the consistent improvements observed across all three benchmarks suggest that the gains are not solely attributable to cryptography-specific learning. That said, we acknowledge that a category-wise breakdown of downstream evaluation results would provide a more precise answer to this question, and we will include this analysis in the final version of the paper to help assess the generality of our approach across different challenge types.

---

> > ### Author Rebuttal · Reviewer_Xiuj · 2026-04-03
> >
> > Thanks for the reply. I will keep my rating unchanged.

---

### Official Review · Reviewer_dfvW · 2026-03-14

**Soundness:** 3
**Presentation:** 3
**Significance:** 3
**Originality:** 3
**Overall Recommendation:** 4
**Confidence:** 4

**Summary:**

The paper proposes CTF-DOJO, a large-scale execution environment built from 658 CTF challenges, together with an automated pipeline (CTF-FORGE) that constructs containerized runtime environments from CTF artifacts. The authors collect execution-verified interaction trajectories from LLM agents solving CTF tasks. These trajectories are then used to fine-tune language models for vulnerability discovery and exploitation. The resulting models demonstrate improved performance on several cybersecurity benchmarks compared with prior approaches.

**Compliance With Llm Reviewing Policy:**

Affirmed.

**Key Questions For Authors:**

The paper is interesting and I appreciate the overall system design. However, I have several questions regarding the rigor of this paper.

For the environment construction component, how does the proposed approach compare with other automated environment construction methods, and which aspects of the pipeline are most critical for achieving the reported environment reconstruction success rates? For example related work such as Repo2Run: Automated Building Executable Environment for Code Repository at Scale.

The validation procedure mainly checks whether containers can be built and whether services respond to requests. How to ensure that the generated environments faithfully reproduce the intended challenge semantics and exploitation paths?

The method uses hints derived from CTF writeups to guide agent exploration. How much these hints contribute to the final performance improvements, for example through ablations or statistics about the information retained in the hints?

**Limitations:**

Yes

**Strengths And Weaknesses:**

# Strengths
1. The paper studies an important and challenging problem. Training LLM agents for vulnerability discovery in realistic execution environments is meaningful for cybersecurity.
2. The scale of the dataset (including both environments and tasks) is valuable and useful for real-world applications.
3. The paper is well written, and the system design and overall pipeline are clearly described.

# Weaknesses

1. The novelty of the CTF environment construction pipeline is not clearly articulated, especially relative to more general automated environment construction approaches. While the automated construction of CTF environments is useful, the paper does not clearly position the proposed pipeline with respect to existing approaches for automated environment reconstruction.
2. The validation of generated environments does not clearly demonstrate that they faithfully reproduce the original challenge semantics or exploitation paths. The current validation procedure mainly checks whether containers can be built and whether services respond to requests. However, these checks may not fully guarantee that the generated environments preserve the intended vulnerability behavior and exploitation paths of the original challenges.
3. The contribution of hints derived from CTF writeups is not fully analyzed. As current hints are derived from writeups that may contain descriptions of vulnerabilities or solution strategies, it is unclear how much of the observed performance improvement is attributable to the hints versus the execution-verified trajectories themselves.

---

> ### Author Rebuttal · Authors · 2026-03-31
>
> Thanks for your valuable review. We address your concerns below:
>
> > For the environment construction component, how does the proposed approach compare with other automated environment construction methods, and which aspects of the pipeline are most critical for achieving the reported environment reconstruction success rates?
>
> We thank the reviewer for pointing out Repo2Run as a relevant comparison. While Repo2Run is an agentic pipeline averaging 29.03 minutes per repository, CTF-Forge adopts a single-turn generation approach that completes environment construction within minutes while maintaining over 98% reliability, making it significantly more time and cost efficient at scale. Beyond efficiency, the construction problem is fundamentally different. Repo2Run targets open-source repositories with ground truth test cases for validation, while CTF-Dojo targets cybersecurity challenges involving diverse and complex scenarios, including container runtimes for agent interaction and compatible web service environments that agents must communicate with during exploitation. A critical aspect enabling our reliability is the diversity of our predefined templates, with more than 10 category-specific templates covering pwn, web, reverse engineering, and cryptography challenges, as well as various file formats. We believe this template diversity, combined with single-turn generation, is the most critical factor in achieving our reported success rates, and will add this comparison to the related work section.
>
> > The validation procedure mainly checks whether containers can be built and whether services respond to requests. How to ensure that the generated environments faithfully reproduce the intended challenge semantics and exploitation paths?
>
> Our automated validation performs two critical checks: whether Docker containers can be successfully built and executed without errors, and whether CTF services respond correctly to network communication on expected ports. Beyond automated checks, we additionally sample 10% of built CTF tasks and manually run executables within each runtime to verify expected behavior including flag submissions. Across three independent runs on all 658 challenges, 98% consistently pass all checks. Regarding exploitation paths, verifying exact paths is fundamentally not possible in the CTF setting, as there are usually multiple valid ways to capture the flag. What matters is that the vulnerability is present and exploitable, and that correct flag submissions are accepted. Our validation procedure, combined with trained agents successfully solving challenges on independent evaluation benchmarks, provides strong evidence that the generated environments faithfully preserve the intended vulnerability behavior.
>
> > The method uses hints derived from CTF writeups to guide agent exploration. How much these hints contribute to the final performance improvements, for example through ablations or statistics about the information retained in the hints?
>
> We provide detailed analysis of hint contributions in Section 4.1. As shown in Table 4, writeup-based hints consistently improve solved tasks across all models and challenge categories. On average, solved challenges increase by 7.4%, from 168 without hints to 217 with hints, with particularly pronounced effects in Crypto, Reverse Engineering, and Miscellaneous categories. Importantly, hints are used only during trajectory collection and are not provided at test time, so performance improvements reflect what the fine-tuned model learned from execution-verified trajectories, not direct hint usage at inference. We acknowledge the valid point about disentangling hint contributions at the training data level. Since hint-guided collection increases successfully solved challenges from 168 to 217, the primary effect is providing a larger and more diverse set of training trajectories, consistent with the scaling law assumption [1] underlying our analysis. A direct ablation comparing models trained on hint-collected versus non-hint-collected trajectories would provide more precise quantification, and we will consider including this in the final version.
>
> [1] Hoffmann, J., Borgeaud, S., Mensch, A., Buchatskaya, E., Cai, T., Rutherford, E., ... & Sifre, L. (2022). An empirical analysis of compute-optimal large language model training. NeurIPS.

---

### Official Review · Reviewer_j13M · 2026-03-22

**Soundness:** 3
**Presentation:** 2
**Significance:** 2
**Originality:** 2
**Overall Recommendation:** 3
**Confidence:** 3

**Summary:**

The paper introduces CTF-DOJO, a collection of executable Caption-The-Flag (CTF) environments as Docker containers. Particularly, the authors select and curate CTF sources, prompt a strong LLM (teacher model) to generate the CTF environments (Docker files and other config files), solve the CTF in the generated environments by means of the strong LLM's agents, and collect the successful traces. Then as a distillation process, these traces are applied to train (fine-tune) weak student models.

**Compliance With Llm Reviewing Policy:**

Affirmed.

**Key Questions For Authors:**

What is the target specific problem addressed in this paper (in one sentence)?

**Limitations:**

Yes

**Strengths And Weaknesses:**

Soundness:
The proposed pipeline and methodology seems to be sound. There are four main steps in generation of the traces:
1. Source collection (mostly manual)
2. Automatic environment generation with CTF-FORGE (mostly automated by the teacher model)
3. Challenge-type-specific setup (mostly automated with manual designed rules and prompts)
4. Validation and challenge solving (mostly automated by the teacher model)

Then the traces are applied to fine-tune weak open-source models. The steps above seem sound.

Presentation:
The presentation is not good.
(1) The problem is not sharply stated. The problem is stated as "To address these limitations, we present CTF-DOJO, the first execution environment ..."; and the limitations are "applying large language model (LLM) agents to CTF challenges ... they fail short when applied to open-source LLMs due to the lack of agentic training data", "synthesizing a large number of long-horizon trajectories from teacher models requires substantial computational resources ...", and "the validity of synthetic trajectories is hard to verify without runtime environments ...". So the problem itself is not sharply/clearly stated.

From the logic chain above, it seems the target problem of the paper is to apply weak open-source LLMs to CTF challenge solving tasks. It is expected that authors state the problem explicitly in one sentence.

(2) Over-claim. The paper uses phrases like "execution-grounded environments remain limited", "the first large-scale executable runtime for training LLMs with verifiable feedback" to imply something, but this is actually over-claim, as their target problem is to apply weak open-source LLMs to CTF challenge solving tasks. What they really did is: automatically building many CTF environments and generating trajectories/traces. The phrases frame a broad problem and there is a conceptual gap between the phrases and the real thing.

It is suspicious that such over-claim/vague presentation of the paper might be "presentation improvement" made by LLM, which typically adds such over-claim and vagueness that does not make much sense.



Significance:
The problem addressed by this paper is actually "how to apply weak open-source LLMs to CTF challenge solving tasks", and its proposed approach is actually "distilling strong teacher LLMs by calling the teacher model to build CTF environments, run CTF tasks, solve CTF challenges, and collect traces, and then fine-tuning student models with the traces".

So this problem is not as significant since strong teacher LLMs have already demonstrated the CTF solving capabilities in prior work, and the approach is just automation of certain engineering work. There is no real research problem here.

Though the paper does not emphasize “teacher–student distillation", but functionally that is what is happening in the proposed approach for the target problem.



Originality:
As mentioned above in the "Significance" section, the approach is just automation of certain engineering work, and there is no real research problem here.

Moreover, most "magic" things are done by the strong teacher models. CTF solving via capable LLM (teacher models) is prior art. The approach's own originality seems limited.

---

> ### Author Rebuttal · Authors · 2026-03-31
>
> Thank you very much for your thorough and thoughtful review. We greatly appreciate the time and effort you have dedicated to providing detailed feedback. Below, we address your comments and concerns point by point.
>
> > What is the target specific problem addressed in this paper (in one sentence)?
>
> Our aim of this work is to enable open-source LLM agents to achieve competitive cybersecurity performance without dependence on costly proprietary systems, by providing scalable execution-verified training environments and trajectories.
>
> > Over-claiming and presention
>
> We sincerely apologize for the current presentation of the Abstract and Introduction sections. The over-claiming statements were made by oversight, and we will revise them carefully in the final version. Specifically, the phrase "the first large-scale executable runtime for training LLMs with verifiable feedback" is indeed too broad and misleading. What we actually contribute is the first large-scale executable runtime specifically designed for cybersecurity agent training, and we will revise this claim accordingly to accurately reflect the scope of our work. We acknowledge that framing this as a general-purpose contribution was incorrect and we apologize for the confusion this may have caused. Similarly, we recognize that the word "pivotal" in describing execution-grounded training signals is stronger than what our evidence strictly supports. Our experiments demonstrate that execution-verified trajectories are effective and data-efficient for cybersecurity tasks, but we do not have sufficient evidence to claim they are pivotal in a broader sense. We will replace this language with more conservative and accurate descriptions that directly reflect our experimental findings. More broadly, we will carefully revise the Introduction and Abstract to ensure that every claim is narrowed to the CTF-specific scope and directly supported by experimental evidence.
>
> > Limited significance and originality
>
> We respectfully disagree that this work is merely "automation of engineering work" or a straightforward teacher-student distillation. We would like to highlight several key aspects that distinguish our work from prior art.
>
> First, while strong teacher LLMs can solve CTF challenges, the critical missing piece is a scalable, verified, and reproducible execution environment that enables training of open-source models. Without CTF-Dojo, there is simply no reliable way to collect execution-verified trajectories at scale for cybersecurity tasks. The environment itself is a non-trivial contribution, as setting up a single CTF runtime can take an experienced practitioner up to an hour, and our pipeline reduces this to minutes while achieving over 98% success rate through manual validation.
>
> Second, our work goes beyond simple distillation by providing three concrete research findings that advance the community's understanding of building effective cybersecurity agents. Specifically, we find that writeups are crucial for training particularly when working with data generated by weak models, that runtime environment augmentation helps models solve more CTF challenges, and that employing diverse teacher LLMs leads to better task diversity and stronger performance. These findings are non-trivial and would not have been possible without the executable runtime environment we provide, as they require careful controlled experiments within a verified execution setting.
>
> Third, the empirical results strongly support the significance of our contribution. Training on just 486 execution-verified trajectories yields up to 11.6% absolute gains over strong baselines, and our best 32B open-source model achieves performance comparable to DeepSeek-V3-0324 and Gemini-2.5-Flash. This demonstrates that our approach effectively closes the gap between open-source and proprietary models in cybersecurity tasks, which we believe is a meaningful and significant contribution to the community. We hope these points help clarify the novelty and significance of our work beyond what may appear as straightforward engineering automation.

---

> > ### Author Rebuttal · Reviewer_j13M · 2026-04-03
> >
> > Thank you for the rebuttal, particularly the stated problem and the plan to change the over-claim presentation.
> >
> > However, the approach is actually still a "teacher-to-student distillation", and all the magic things, including "build CTF environments, run CTF tasks, solve CTF challenges, and collect traces", are all done by the strong teacher model. The research contribution is just a simple fine-tuning, and the problem does not seem to be a research problem.

---

> > > ### Author Response · Authors · 2026-04-04
> > >
> > > We appreciate the reviewer's follow-up and would like to clarify **an important misunderstanding**. The core contribution of this work is the infrastructure itself, namely a scalable, verified, and reproducible execution environment for **cybersecurity agent training**, rather than the specific training recipe applied on top of it. Fine-tuning on execution-verified trajectories is one instantiation of how CTF-Dojo can be used, chosen primarily to validate that the environments are correctly constructed and that the collected trajectories are of sufficient quality to be useful for training.
> > >
> > > CTF-Dojo opens the door to a much broader range of research directions that go well beyond teacher-to-student distillation. Most notably, the verified execution environments we provide are precisely the ingredient needed to enable reinforcement learning for cybersecurity agents, where reward signals are derived directly from whether the agent successfully captures the flag in a real runtime. This is analogous to how verified execution environments in mathematics and coding have been fundamental enablers of frontier model capabilities in those domains. To our knowledge, no such large-scale verified execution environment has previously existed for cybersecurity tasks, and we believe CTF-Dojo will serve as a critical foundation for future work in this direction, including but not limited to RL-based agent training, automated vulnerability discovery, and scalable security benchmarking. We will make this broader vision more explicit in the final version of the paper.

---

### Decision · Program_Chairs · 2026-04-30

**Decision:**

Accept (regular)

**Comment:**

This paper presents CTF-Dojo, a scalable pipeline for constructing 658 containerized CTF challenges, demonstrating significant downstream gains (up to 11.6% Pass@1) when used for agent fine-tuning.

The reviewers (1 Accept, 2 Weak Accepts, 1 Weak Reject) unanimously praised the robustness of this infrastructure—achieving ~98% reliability—which fills a critical gap in open-source cybersecurity agent research. While I agree with the concerns regarding limited methodological novelty and initially over-claimed abstract, these are substantially outweighed by the difficulty and value of building a reproducible execution environment in this blank area. The authors have explicitly acknowledged and agreed to rectify the over-claiming. Therefore, I recommend Weak Accept.